# Classification of Clinically Significant Prostate Cancer on Multi-Parametric MRI: A Validation Study Comparing Deep Learning and Radiomics

**DOI:** 10.3390/cancers14010012

**Published:** 2021-12-21

**Authors:** Jose M. Castillo T., Muhammad Arif, Martijn P. A. Starmans, Wiro J. Niessen, Chris H. Bangma, Ivo G. Schoots, Jifke F. Veenland

**Affiliations:** 1Department of Radiology and Nuclear Medicine, Erasmus MC, 3015 GD Rotterdam, The Netherlands; j.castillotovar@erasmusmc.nl (J.M.C.T.); a.muhammad@erasmusmc.nl (M.A.); m.starmans@erasmusmc.nl (M.P.A.S.); w.niessen@erasmusmc.nl (W.J.N.); i.schoots@erasmusmc.nl (I.G.S.); 2Faculty of Applied Sciences, Delft University of Technology, Lorentzweg 1, 2628 CJ Delft, The Netherlands; 3Department of Urology, Erasmus MC, 3015 GD Rotterdam, The Netherlands; c.h.bangma@erasmusmc.nl; 4Department of Medical Informatics, Erasmus MC, 3015 GD Rotterdam, The Netherlands

**Keywords:** prostate carcinoma, clinically significant, radiomics, machine learning, deep learning, comparison, mpMRI, classification, model, prediction, Gleason score

## Abstract

**Simple Summary:**

Computer-aided diagnosis systems to improve significant prostate cancer (PCa) diagnoses are being reported in the literature. These methods are based on either deep learning or radiomics. However, there is a lack of scientific evidence comparing these methods on the same external validation sets. The aim of our study was to compare the performance of a deep-learning model with the performance of a radiomics model for significant-PCa diagnosis on various cohorts. We collected multiparametric magnetic resonance images and pathology data from four patient cohorts (644 patients in total). One of the cohorts was used to develop a deep-learning model and a radiomics model. Both models were tested on the three remaining cohorts. The comparison shows that whereas the performance of the deep-learning model was higher on the training cohort, the radiomics model outperformed the deep-learning model in all the testing cohorts, making it a more accurate tool with which to detect clinically significant prostate cancer.

**Abstract:**

The computer-aided analysis of prostate multiparametric MRI (mpMRI) could improve significant-prostate-cancer (PCa) detection. Various deep-learning- and radiomics-based methods for significant-PCa segmentation or classification have been reported in the literature. To be able to assess the generalizability of the performance of these methods, using various external data sets is crucial. While both deep-learning and radiomics approaches have been compared based on the same data set of one center, the comparison of the performances of both approaches on various data sets from different centers and different scanners is lacking. The goal of this study was to compare the performance of a deep-learning model with the performance of a radiomics model for the significant-PCa diagnosis of the cohorts of various patients. We included the data from two consecutive patient cohorts from our own center (*n* = 371 patients), and two external sets of which one was a publicly available patient cohort (*n* = 195 patients) and the other contained data from patients from two hospitals (*n* = 79 patients). Using multiparametric MRI (mpMRI), the radiologist tumor delineations and pathology reports were collected for all patients. During training, one of our patient cohorts (*n* = 271 patients) was used for both the deep-learning- and radiomics-model development, and the three remaining cohorts (*n* = 374 patients) were kept as unseen test sets. The performances of the models were assessed in terms of their area under the receiver-operating-characteristic curve (AUC). Whereas the internal cross-validation showed a higher AUC for the deep-learning approach, the radiomics model obtained AUCs of 0.88, 0.91 and 0.65 on the independent test sets compared to AUCs of 0.70, 0.73 and 0.44 for the deep-learning model. Our radiomics model that was based on delineated regions resulted in a more accurate tool for significant-PCa classification in the three unseen test sets when compared to a fully automated deep-learning model.

## 1. Introduction

Prostate cancer (PCa) diagnosis using prostate-specific antigen (PSA) and transperineal ultrasound-guided biopsies combined with multiparametric magnetic resonance imaging (mpMRI) is recommended by the European guidelines [1] and common practice. In the literature, it was shown that the multiparametric MRI (mpMRI)-targeted biopsy can non-invasively and more accurately characterize PCa lesions as compared to the standard systematic transrectal ultrasound-guided (TRUS) biopsy [2,3,4]. Furthermore, the combination of the MRI-targeted biopsy with the systematic biopsy increases the overall accuracy of PCa diagnoses [5].

The mpMRI interpretation of the prostate was standardized by the Prostate Imaging Reporting and Data System (PI-RADS) v2. However, the visual interpretation by radiologists can still lead to the under-diagnosis of clinically significant PCa and the over-diagnosis of insignificant PCa [6].

A computer-aided quantitative analysis of prostate mpMRI could improve PCa detection and may help in the standardization of mpMRI interpretation [7]. Ultimately, it may contribute to improving the diagnostic chain [8], thereby reducing over- and under-diagnoses in prostate-cancer management [9]. Several methods for significant-PCa segmentation [10,11,12] or classification [13,14,15] using deep-learning networks or radiomics approaches have been reported in the literature. Both approaches offer different capabilities, challenges and difficulties [16]. Comparing the performance of both approaches based on the scientific literature can be difficult.

First, because direct comparison of the methods is often not specifically addressed or examined [17]. Second, in general, these developed methods are cohort dependent, and their performance may substantially decrease for independent test data. Differences in the data sets used, such as the selected patient population, the data-set size, the scanner type and manufacturer, the MRI protocols, the image quality, the radiologist’s delineations, the pathology material (biopsies or radical prostatectomies), and the pathology reports may all substantially influence the model performance [18]. Several approaches (deep learning and radiomics) have been compared based on the same data set [19]; however, the comparison of the performances of both approaches on multiple data sets obtained at different institutions is lacking.

Therefore, the aim of this study was to perform a comparison study of a deep-learning method for significant-PCa segmentation and a radiomics-based significant-PCa-classification method on various patient data sets.

## 2. Materials and Methods

In this study, we compared the performance of a recently developed deep-learning network for significant-PCa segmentation [10] with a radiomics approach based on regions of interest (ROIs) that were delineated by a radiologist. Both models were trained on the same data set. Three data sets, representing various scanners, various patient populations from different hospitals, and two different ground-truth methods were used to compare the performances of both methods. PCa with an ISUP grade ≥ 2 (Gleason Score (GS) 3 + 4 and higher) was classified as significant [20]

### 2.1. Patient Data Sets

This study included four patient cohorts: an Active-Surveillance cohort (AS), a cohort of men with previous negative systematic biopsies (Prodrome), the cohort included in the Prostate Cancer Molecular Medicine study (PCMM) and the ProstateX challenge cohort, as illustrated in Figure 1. The cohort of men participating in the Active-Surveillance study and the cohort of the previous-negative-biopsy men were acquired from prospective studies of the Erasmus MC in Rotterdam, the Netherlands [21]. The usage of the data for this study was HIPAA compliant and written informed consent with a guarantee of confidentiality was obtained from the participants. The ProstateX data were publicly available and were provided for the SPIE-AAPM-NCI ProstateX challenge [19,22,23]. The PCMM data set was acquired retrospectively from two external healthcare centers in the Netherlands [18], the data usage of this study was approved by the medical ethics review committee of the Erasmus MC under the number NL32105.078.10.

In total, 644 patients from all four cohorts were included in this study. The selected patients were divided into a training set (Active-Surveillance cohort) and test sets (Prodrome, ProstateX and PCMM cohorts). The general patient characteristics (prostate volume, age and prostate-specific antigen) of the sets are listed in Table 1. Prostate volume was measured on T2-weighted (T2w) images using the ellipsoid formulation [24].

For the Active Surveillance cohort, initially, 377 consecutive patients with low-risk PCa (defined as International Society of Urological Pathology “ISUP” grade 1) were prospectively enrolled in our in-house database from 2016 to 2019. All participants received a multi-parametric MRI and targeted biopsies of visible, suspicious (PI-RADS ≥ 3) lesions at the baseline (3 months after diagnosis). A detailed description of the data set has been published [25]. Patients who refused, who had no biopsy procedure, or whose MRI scans had artifacts were excluded from the study. Since the systematic-biopsy locations were not available, patients in whom significant PCa was found based only on systematic biopsies were also excluded. The remaining cohort (*n* = 271) was divided into two groups based on the pathology findings (Figure 1a). Some of the patients (*n* = 55) with an ISUP grade = 1 did not have an identifiable lesion on MRI, therefore no targeted biopsy was performed, and the systematic biopsy was negative.

The Prodrome cohort contained 111 consecutive patients with prior negative biopsies who were prospectively enrolled in our in-house database from 2017 to 2019. For each patient, mpMRI with blinded systematic biopsies and MRI-targeted biopsies of visible, suspicious (PI-RADS ≥ 3) lesions were performed. Patients who refused the biopsy procedure, or whose MRI scans had artifacts were excluded from the study. Patients who were found to have significant PCa based only on systematic biopsies were also excluded since the exact locations of the systematic biopsies was unknown. The remaining cohort (*n* = 100) was divided into two groups based on the pathology findings (Figure 1b).

The ProstateX cohort comprises 204 patients and included suspicious-lesion coordinates and targeted-biopsy-based histopathological findings. Nine patients were excluded due to image artifacts, registration and missing DWI images at b800. The remaining cohort (*n* = 195) was divided into two groups based on the pathology findings (Figure 1c).

The PCMM cohort consists of 107 patients who were enrolled from 2011 to 2014 and included lesion segmentations based on delineations made by a pathologist on prostatectomy specimen photos. The MR images of these patients were correlated with the prostatectomy photos using manual registration [18]. Twenty-nine patients were excluded due to image artifacts or low resolution (Figure 1d).

For each patient, two MRI sequences, axial T2w and diffusion-weighted images (DWIs) with their apparent-diffusion-coefficient maps (ADC) were selected. For the AS, Prodrome and ProstateX cohorts, the histopathology data from the MRI-targeted biopsies were considered as reference standards. In the case of the PCMM data set, the ground truth was obtained from pathology reports after the prostatectomy.

### 2.2. MR Imaging and Pre-Processing

For the Active Surveillance and Prodrome cohorts, the MRI scans were performed on a 3T system (Discovery MR750, General Electric Healthcare, Chicago, IL USA), according to the PI-RADS v2 guidelines [26]. The T2-weighted imaging (T2w) diffusion-weighted imaging (DWI, b-values 50, 400 and 800) were acquired using a 32-channel pelvic phased-array coil with 0.371 × 0.371 × 3.3 mm^3^ resolution. Apparent-diffusion-coefficient (ADC) maps were constructed using scanner software. All MR images were reviewed by a urogenital radiologist with over 6 years of prostate MRI experience. Individual lesions with a PI-RADS score ≥ 3 were defined as suspicious and delineated in the T2w images. The MRI-and-transrectal-ultrasound (TRUS)-fusion technique was used (UroStation™, Koelis, France) to perform targeted biopsies with a maximum of 4 cores under ultrasound guidance. One expert uropathologist reviewed the biopsy specimens according to the ISUP 2014 modified Gleason Score [27]. For every patient, binary masks were generated for our experiments based on the delineations on the T2w images with biopsy-proven significant PCa (ISUP grade ≥ 2). The DWIs with ADC values were manually and rigidly co-registered to the T2w images for every patient. Furthermore, the 3D images were cropped to the whole prostate region of interest with dimensions 128 × 192 × 24 voxels along the x, y and z directions.

For the ProstateX cohort, MRI scans were acquired on one of two 3T MR systems (MAGNETRON Trio and Skyra, Siemens Medical Systems, Erlangen, Germany). The MRI protocol included T2w images acquired with 0.5 mm 2D resolution and 3.6 mm slice thickness, DWI (b-values 40, 400 and 800) series acquired using single-shot-echo-planer imaging with 2 mm 2D resolution and 3.6 mm slice thickness, and ADC maps constructed using scanner software. Since the voxel sizes in the data from the ProstateX cohort varied, all images were resampled to a uniform voxel spacing of 0.371 × 0.371 × 3.3 mm^3^, which was the same as the Active Surveillance and Prodrome cohorts. The DWIs with ADC values were manually and rigidly co-registered to the T2w images for every patient and cropped to the whole prostate region of interest with dimensions 128 × 192 × 24 voxels along the x, y and z directions. For every patient, significant PCa (ISUP grade 00) was delineated based on each given lesion’s coordinates by one investigator with 6 months experience under supervision and in consensus with a urogenital radiologist with over 6 years of prostate MRI experience.

The MRI scans of the PCMM data set were obtained from three different 3T MR systems. Two were Siemens Medical system models (MAGNETRON Trio and Skyra) with the same characteristics as the models described for the ProstateX cohort. The third scanner model was from Philips (Achieva). The MRI protocol included T2w images with 0.27 mm 2D resolution and 3.00 mm slice thickness, DWIs (b-values 100, 300, 450, 600, 750) with 1.03 mm 2D resolution and 3 mm slice thickness, and ADC maps obtained from the scanner. The images from the Philips MRI were acquired using an endorectal coil. All the original images were sampled to match the voxel spacing of the Active Surveillance cohort and cropped from the central region of the image having dimensions of 128 × 192 × 24 voxels along the x, y and z directions. The lesion delineations were based on the manual registration with the pathology specimens delineated by an expert pathologist.

All the T2-weighted images in this study were pre-processed using z-scoring for pixel-intensity normalization. The MRI scans from the patient populations included in this study were from 2013 to 2019. Therefore, high b-values images (>1400) were acquired for only a portion of the patient cohorts. Since we did not want to mix the artificially extrapolated and the acquired high b-value images, we chose to focus on the raw data.

### 2.3. Development of the Models

For this experiment, both models were trained on images from the Active Surveillance cohort. Whereas the deep-learning network was trained to identify significant PCa in images of the whole prostate, the radiomics model was trained to identify significant PCa based on the ROIs that were delineated by the radiologist.

#### 2.3.1. Deep-Learning Network

A fully convolutional neural network (CNN), as described in a recently published previous work was used [10]. Three 3D MR images (T2w, DWI (b-value closest to 800) with corresponding ADC map) were used as inputs for the CNN. Each sequence was considered as a separate input channel. The network contained twelve single 3D convolution layers. Batch normalization (BN) was added after each 3D convolution to improve the convergence speed during training. In the final layer, a 3D convolution having 1 × 1 × 1 kernel size was used to map the computed features to the predicted significant-PCa segmentation.

The network was trained in Python (version 3.5.3) using Keras (version 2.0.2) with Tensor Flow (version 1.0.1) as the backend. During training and prediction, a GeForce GTX TITAN Xp GPU (NVIDIA Corporation) was used. The loss function during training was the binary cross-entropy metric and was optimized using an Adam optimizer with a learning rate of 0.01. For better generalization, data augmentation was implemented in all images during training, which included rotation (0–50, along *x*,*y*,*z*-axes) and shearing (along *x*,*y*,*z*-axes) with rigid transformation and 50% probability. The output of the trained network was the binary segmentation (voxel values from 0 to 1) of clinically significant PCa lesions.

#### 2.3.2. Radiomics Model Development

The radiomics model that was used to classify ROIs as significant versus insignificant PCa was developed with the open-source Workflow for Optimal Radiomics Classification (WORC) package for python [18,28]. Similar to the deep-learning network, the inputs for WORC were the T2w MRIs, the DWIs (b-value closest to 800) and ADC maps. Additionally, the ROIs of the lesion delineations were provided. Within the ROIs, 564 radiomic features quantifying intensity, shape, texture and orientation were extracted from the two MR images and the ADC map.

WORC performs an automated search amongst a wide variety of algorithms and their corresponding parameters in order to determine the optimal combination that maximizes the prediction performance on the training set. During each iteration, WORC generates 1000 workflows by using different combinations of methods and parameters. The internal evaluation of the model was performed by using a 100x random-split cross-validation.

At the end of each cross-validation, the 100 best-performing solutions were combined in an ensemble as a single classification model by averaging their probability predictions. The details regarding the feature computation, model selection and optimization can be found in Appendix A and Appendix B, respectively.

### 2.4. Methods Performance Comparison

The performances of both methods were evaluated using receiver-operating-characteristic (ROC) curves, accuracy and F1-scores that were computed at the patient level both by internal cross-validation and external validation. Internal validation was performed using a 3-fold random cross-validation, which randomly split the training set with ISUP ≥2 patients into three separate folds (fold 1 contained 39 patients, fold 2 contained 38 patients and fold 3 contained 39 patients). In each iteration, the models were trained on two of the three folds, and were evaluated on the third fold and an independent test set containing 155 patients with ISUP grades ≤ 1, more details can be found in [10]. These splits were kept the same for the training of both models. The average over these three iterations is reported.

To compare the performance with clinical practice, the sensitivity and specificity of the visual scoring by the experienced radiologist is indicated (when available).

## 3. Results

### 3.1. Internal Cross-Validation

In Figure 2, the ROC curve of the deep-learning model (blue line) and the radiomics model (orange line) as computed by the internal cross-validation of the training set can be seen. Table 2 depicts the accuracy, sensitivity, specificity and F1-score obtained from this experiment. Overall, it can be seen that deep learning performed better on the training set compared to the radiomics model.

### 3.2. External-Validation

Figure 3a–c depict the ROC curves of the deep-learning and radiomics models in the Prodrome, ProstateX and PCMM cohorts. Furthermore, the rest of the evaluation metrics can be found in Table 2. In the Prodrome cohort, the deep-learning model obtained an AUC of 0.70 versus 0.88 for the radiomics model, whereas in the ProstateX and PCMM cohorts the AUCs were 0.73 and 0.44 for the deep-learning model and 0.91 and 0.65 for the radiomics model, respectively.

To illustrate the deep-learning-segmentation method, examples (true positive, false negative and false positive) are shown in Figure 4. In the true-positive example (Figure 4A) the network has successfully segmented both significant-PCa lesions as delineated by the radiologist and proven by targeted biopsy. In some cases, PCa segmentation was unsuccessful, leading to a false negative (Figure 4B). In the false-positive example (Figure 4C), the deep-learning network segments a lesion in the anterior stroma zone; however, the targeted biopsy found insignificant PCa (ISUP grade 1, GS 3 + 3 = 6).

## 4. Discussion

The detection of significant PCa utilizing CAD systems that were based either on deep learning or radiomics has generated many papers in the scientific literature [29,30,31]. Both approaches offer different capabilities, challenges and difficulties [16]. Whereas both approaches have been compared based on the same data set [19], the direct comparison of these approaches when tested on the same unseen data is lacking in the current scientific literature. To our knowledge, this is the first study comparing a fully automated deep-learning model for significant-PCa segmentation with a radiomics approach based on segmentations that were performed by a radiologist. This study used a large patient cohort compared to the average sizes used in similar studies [29] (avg = 127 patients), and takes into account data from various patient risk groups, various MRI scanners in various hospitals, and ground truths obtained from both biopsies and prostatectomies.

The deep-learning and radiomics models obtained similar ROC curves when validated by internal cross-validation on the training set. However, when validated by external sets, the comparison showed that the radiomics model had higher AUC values than the automated deep-learning model in all data sets.

Nevertheless, when comparing, some considerations should be mentioned. First, there was a tissue-volume difference; the deep-learning model was trained to perform the segmentation of significant prostate cancer in the whole prostate, whereas the radiomics model classified the ROIs that were delineated by the radiologist. For the delineated ROIs, the ground truth was known, but when the deep-learning model segmented a lesion where no biopsy was taken, this lesion was considered to be a false positive, while in fact we did not know the ground truth.

Second, the better performance of the radiomics model could also be attributed to the set of features used by the model, which was a generic set of features that have been used for other applications [15,32]. Furthermore, the deep-learning model automatically designed features specific to the training data and it is possible that this set of features was too specific, causing the generalization issues across the testing sets. Third, the radiomics model used an ensemble of many machine-learning models. Utilizing ensembles might have granted the radiomics model a generalization ability that the deep-learning model lacked by being a model based on a single CNN [33]. This difficulty to generalize was evident in the test sets that used a different ground truth, different scanners and older acquisition protocols. Fourth, the deep-learning approach required a larger number of training examples than the radiomics approach [34]. Therefore, the limited number of patients available for training could have reduced the performance of the deep-learning model compared to the radiomics performance.

An interesting aspect of the better performance of the radiomics model is revealed when comparing it to the performance of a radiologist. For the Prodrome and PCMM cohorts, the combination of the radiologist’s delineations with the radiomics model allowed for the preservation of the radiologist’s high sensitivity while increasing the specificity for clinically significant PCa lesions. Adding a radiomics model to the workflow could mean a more accurate patient selection for the biopsy procedures and a reduction in overdiagnoses.

In the ProstateX challenge, several methods were compared for clinically significant-PCa classification [19]. However, the details regarding the training setup and methods were not described, thereby prohibiting a comparison with our results. The results obtained from other tumors [35,36,37] that were all evaluated on their own single internal validation set show that deep-learning methods outperformed radiomics, which corresponds to our findings. One study [38] that performed external validation found that radiomics obtained better results than deep learning, which also corresponds to our results.

There are some limitations to this study. For instance, the clinical variables such as age, race, and PSA level were not available for some of the patient cohorts and therefore could not be included in the models. This information is frequently taken into account by clinicians as risks factors for aggressive prostate cancer [39]. Secondly, co-existing benign prostatic diseases that can mimic PCa were not taken into account in our experiments, since no ground truth was available for these diseases. Furthermore, part of our data is related to biopsy results, which reflect the diagnostic outcome. However, biopsy results can be downgraded or upgraded in radical prostatectomy specimens, which may hamper the interpretation of the results.

Another limitation was that the delineations in this study were carried out by a single clinician; therefore, we were not able to study the effect on the feature computation or the performance with multiple evaluators. Lastly, this was a retrospective study, which precluded the comparison of the models while they were being used by a clinician.

PCa diagnosis can be improved using CAD systems that are based on radiomics or deep learning. However, there is no evidence of these methods being used in clinics in prospective studies [29]. Hence, future research should focus on studying the impact of the radiologist’s decision to incorporate either deep-learning or radiomics models in the workflow.

## 5. Conclusions

Both deep-learning and radiomics methods provide capabilities to support PCa diagnoses. In this study, a radiomics model that was trained on segmentations provided by a radiologist resulted in a more accurate and generalizable tool for significant-PCa classification compared to a fully automated deep-learning model for significant-PCa segmentation.

## Figures and Tables

**Figure 1 cancers-14-00012-f001:**
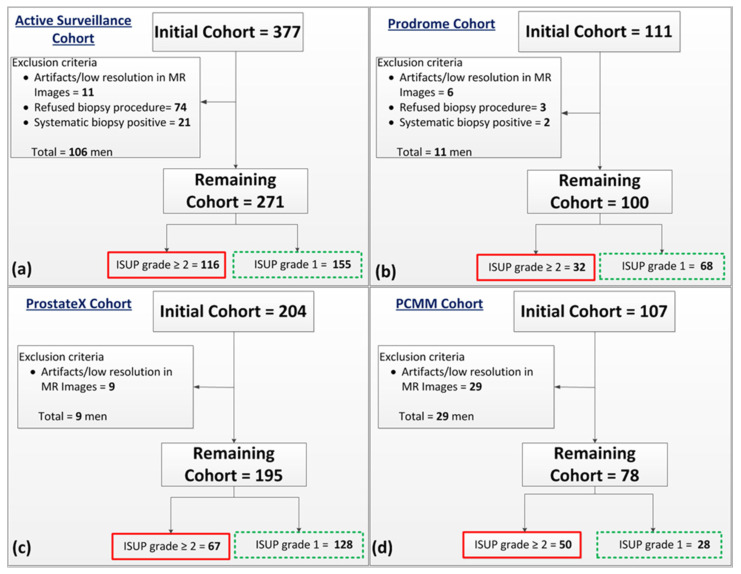
Flow diagram of patient exclusion and inclusion of the four cohorts used in this study: (**a**) Active Surveillance, (**b**) Prodrome, (**c**) ProstateX, and (**d**) PCMM. **ISUP**: International Society of Urological Pathology.

**Figure 2 cancers-14-00012-f002:**
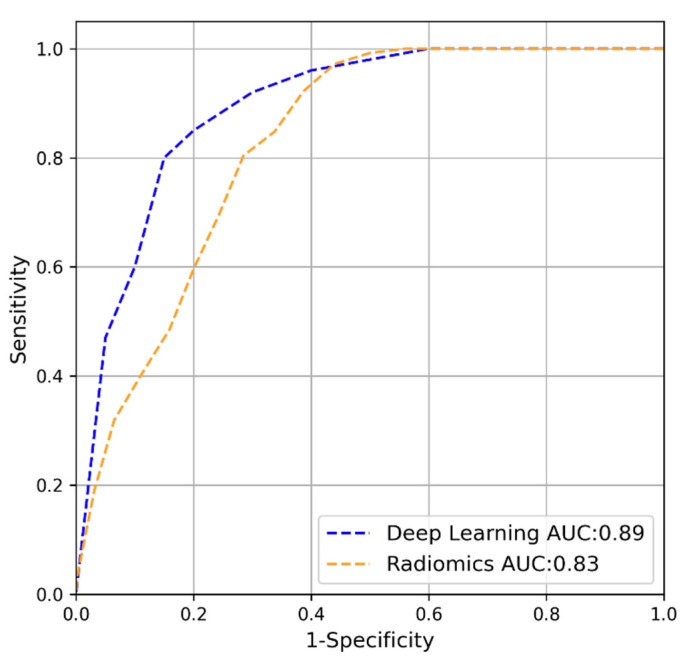
ROC curves of the deep-learning (blue) and radiomics (orange) models on the interval validation on Active Surveillance.

**Figure 3 cancers-14-00012-f003:**
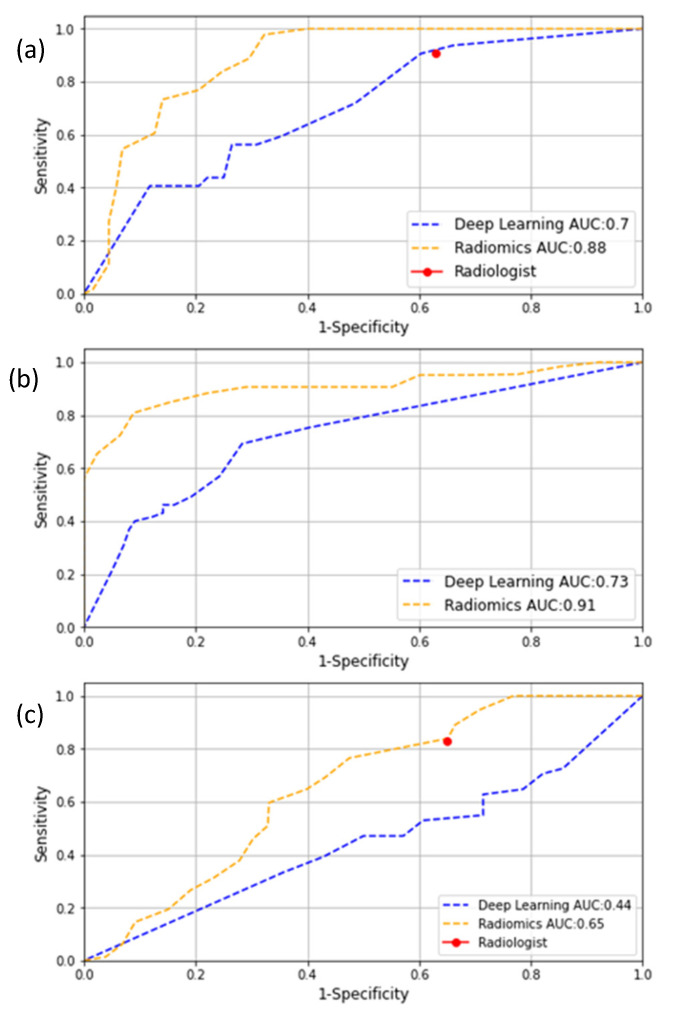
ROC curves of the deep-learning (blue) and radiomics (orange) models when evaluated on the test sets: (**a**) Prodrome, (**b**) ProstateX and (**c**) PCMM.

**Figure 4 cancers-14-00012-f004:**
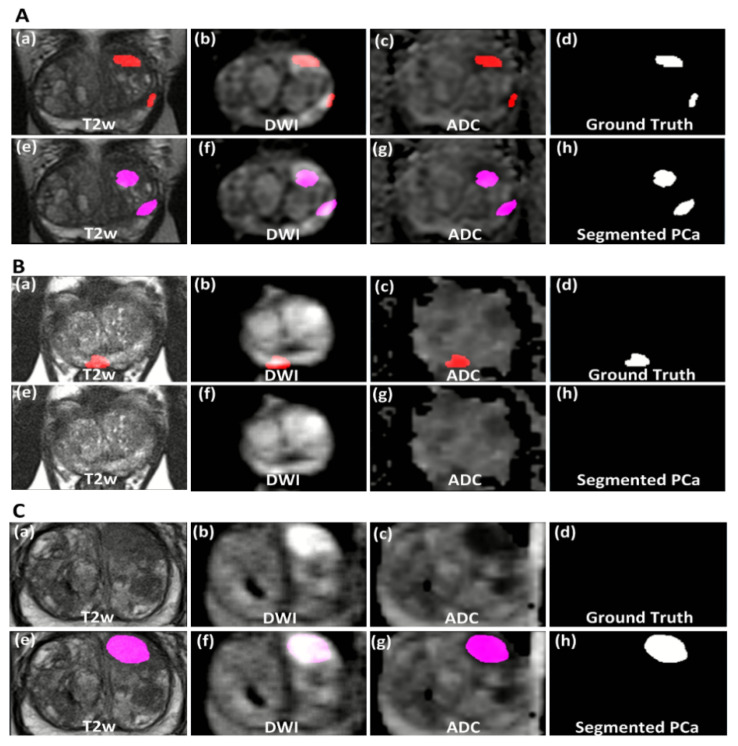
All images show the same axial slice as 2D view of mpMR images (**a**,**e** T2w images; **b**,**f** DWI b800; **c**,**g** ADC map) of the prostate with the reference ground truth (**d**) and the segmented PCa lesion by the deep-learning model (**h**) (**A**) Example of a true positive case (PSA = 17.6; prostate volume = 46 cc; ISUP grade = 2 (up) and 3 (down)). The ground truth is shown in overlay (red) as delineated by the radiologist and proven by targeted biopsy as significant PCa. The segmented significant-PCa lesion by the deep-learning model is shown in overlay (pink). (**B**) Example of a false negative case (ISUP grade = 2). The ground truth is shown in overlay (red) as delineated by the radiologist and proven by targeted biopsy as significant PCa. The deep-learning model has not segmented any PCa lesion. (**C**) Example of a false positive case (ISUP grade = 1). The images show no delineation due to the absence of significant PCa, the region delineated by the radiologist (not shown) proved by targeted biopsy as insignificant PCa. The lesion segmented incorrectly by the deep-learning model is shown in overlay (pink).

**Table 1 cancers-14-00012-t001:** Clinical characteristics of the patients of the four cohorts (Active Surveillance, Prodrome, ProstateX and PCMM) included in this study. Tumorr volume values are presented as median (interquartile range). **PZ**: peripheral zone. **TZ**: transition zone. **ISUP**: International Society of Urological Pathology. **GSA:** Gleason Score. **PSA**: Prostate Specific Antigen. **NA**: not available. (*****) The ISUP grade per lesion was not available for ProstateX challenge, the ground truth provided for this set indicated whether the lesion had ISUP grade ≥ 1.

	Training Cohort	Testing Cohort		
Patient Cohort	Active Surveillance	Prodrome	ProstateX *	PCMM
Total Number of patients	271	100	195	78
Patients with a lesion ISUP grade = 1	155	68	128	28
Patients with a lesion ISUP grade ≥ 2	116	32	67	50
**Total number of lesions**	233	104	328	156
ISUP grade 1	100	52	254	77
ISUP grade ≥ 2	133	52	74	79
ISUP grade 2	124	45	NA	68
ISUP grade 3	3	6	NA	8
ISUP grade 4 & 5	6	1	NA	3
Lesions in PZ	150	60	191	104
Lesions in TZ	33	41	82	49
Lesions in other zones (central, anterior stroma)	38	3	55	3
Lesion volume (mL)	0.3(0.2–0.8)	0.61 (0.3–1.0)	1.42 (1.4–3.2)	0.80 (0.2–1.1)
Prostate Volume(mL)	43.1 (30.5–76.2)	50. (33–67)	NA	NA
Age (year)	67 ± 7	68 ± 4	NA	NA
PSA(mean ± std ng/mL)	10 ± 6	12 ± 4	NA	9 ± 7

**Table 2 cancers-14-00012-t002:** Deep-learning- and radiomics-model performances on the training set (Active Surveillance) and on the external sets (Prodrome, ProstateX and PCMM). **DL**: deep learning. **AUC**: Area under the curve.

	Active Surveillance		Prodrome		ProstateX		PCMM	
Metrics	DL	Radiomics	DL	Radiomics	DL	Radiomics	DL	Radiomics
AUC	0.89	0.83	0.70	0.88	0.73	0.91	0.44	0.65
Accuracy	0.76	0.63	0.58	0.78	0.71	0.85	0.52	0.55
Sensitivity	0.85	1.00	0.72	1.00	0.70	0.72	0.70	0.44
Specificity	0.52	0.54	0.51	0.68	0.71	0.94	0.18	0.71
F1-score	0.74	0.66	0.52	0.78	0.65	0.85	0.66	0.55

## Data Availability

Publicly available datasets were analyzed in this study. This data can be found here: https://wiki.cancerimagingarchive.net/pages/viewpage.action?pageId=23691656 (accessed on 18 December 2021).

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
