# Peer review of "Classification of Clinically Significant Prostate Cancer on Multi-Parametric MRI: A Validation Study Comparing Deep Learning and Radiomics"

_cancers, 2021, doi:10.3390/cancers14010012_

Round 1
Reviewer 1 Report
This study aims to compare the performance of a deep learning model with the performance of a radiomics model for a significant prostate cancer diagnosis.
I believe that the study has sufficient merit to be considered for publication on Cancers, although major revisions are required.
Discussion. I believe that authors should include more information also in view of current literature. This interesting paper deserve to read about: https://pubmed.ncbi.nlm.nih.gov/34576134/; DOI: 10.3390/ijms22189971
Author Response
Answer: Thank you for your comment, however we do not completely understand what type of information you are requesting to be included or where to include it.
We added, to give a broader overview of the literature, in the discussion, two systematic reviews(suggested literature included) on the application of radiomics in PCa(Line 301).
Reviewer 2 Report
valuable research
Author Response
Answer: Thanks for your comments.
Reviewer 3 Report
The article was properly revised to the previous review comments from me.
Just one more question remained.
Is the number of ISUP grade ≥ 2 in Prodrome cohort 52 rather than 53?
In my calculation, 45+6+1 = 52. Please verify this.
Thank you for your revision
Author Response
Answer: Thank you for close reading the manuscript. The discrepancy was fixed on Table 1.
Round 2
Reviewer 1 Report
Materials and Methods. Authors should enlighten the baseline characteristics (bc) of PC patients included in the studies (prostate volume); in this regard this interesting and novel paper (doi: 10.1159/000516681; PMID: 34247169) shows how prostate volume is a useful tool in risk stratification, diagnosis, and follow-up of prostate cancer.
Author Response
Answer: Thank you for the comment. In line with the comment on the baseline characteristics, we added in the methods section to our manuscript how the prostate volume was measured in our data referring to the PI-RADS V2.1 standard.
We want to stress that the prostate volume was not a variable included in our experiments, it is only provided to characterize the study populations. Whether or not the prostate volume is a useful tool for stratification was not part of our research hypothesis. Moreover, the recommended paper compares different methods to estimate prostate volume but does not show the usefulness of prostate volume in risk stratification.
We think that the recommended paper is beyond the scope of our research.
We added the following text in line 108:
"General patient characteristics (prostate volume, age and prostate specific antigen) of the sets are listed in Table 1. Prostate volume was measured on T2-weighted (T2w) images using the ellipsoid formulation[24]."
This manuscript is a resubmission of an earlier submission. The following is a list of the peer review reports and author responses from that submission.
Round 1
Reviewer 1 Report
Overall: This is an interesting study about external validation of DL and radiomics of prostate MRI. Prostate cancer is very challenging subject for these techniques, because the mass is not readily delineated in MRI.
Pros of this study:
External validations to three different cohort
Modest number of the subjects in the cohort
Cons of this study:
Based on biopsy results other than whole mount histopathology
Comparison of DL/radiomics is not head to head
1. Title
Prostate Cancer Classification on Multi-parametric MRI: a Validation Study comparing Deep Learning and Radiomics.
--> In this study, DL was used to segment clinically significant cancer and radiomics was used to classify cancer with ISUP grade 1 vs ISUP grade 2. I think ‘classification’ is too broad terminology for this task, because at least cancer with ISUP grade 3,4,5 should be differently evaluated, but missing in this study about these high grade cancers.
2. Materials and methods
A. Questions about the cohort: (mainly in Table 1)
2-1. Training cohort had 271 patients with 221 lesions. Why the number of the lesions are smaller than the number of the patients?
2-2. In ProstateX testing cohort, total number of lesions was 328. The number of ISUP Gr 1 was 254. Grades of the rest 74 lesions are missing. Explain this.
B. MR imaging and pre-processing
2-3. in Line 151. The authors used DWI with b-value 50,400 and 800. According to PIRADS V2, there should be a very high b-value image over 1400 is required for the scoring. Did you perform this high b-value image or not? If you did, explain why this images were not used.
C. Deep learning model
2-4. Please describe how many patients were allocated in the training, validation and test group in internal validation group.
3. Discussion
3-1. This study is based on biopsy result. However, ISUP grade from the biopsy can be upgraded or downgraded after prostatectomy in many patients. This should be mentioned in the limitation section with appropriate references.
4. If there was a lesion of ISUP Gr 1 with expected tumor volume over 0.5 mL in DL model, did you classify this as clinical significant (CS) or insignificant cancer? I think it should be classified as a CS cancer, according to Epstein Criteria. However, but should be graded as an ISUP Gr 1 by radiomics. How did you deal with this contradiction?
Thank you.
Reviewer 2 Report
This study aims to compare the performance of a deep learning model with the performance of a radiomics model for a significant prostate cancer diagnosis. The authors included 644 patients divided into four cohorts: an Active Surveillance cohort, used as a training set and the three remaining cohorts (Prodrome, ProstateX and PCMM), used as a test set. The comparison shows that the radiomics model is a more accurate tool to detect clinical significant prostate cancer than the deep learning model.
I believe that the study has sufficient merit to be considered for publication on Cancers, although major revisions are required.
Introduction. The definition of clinically significant and insignificant prostate cancer. It would appropriate to include the ISUP Grade Group System, repeatedly mentioned in the work and not fully described. In lines 53-55 authors must better discuss the accuracy of the mp-MRI targeted biopsy, underlining that a combined approach of fusion targeted and systematic biopsy could further increase the overall accuracy in prostate cancer diagnosis, especially in biopsy-naïve patients.
Materials and Methods. In line 122 there is a discrepancy between the number of patients reported in the text and that in Figure 1 (101 in the text, 100 in Figure 1). In Table 1 the tumour volume values and the locations of the lesions are shown. The use of Computer aided diagnosis systems can be justified in relation to these parameters? (e.g. in case of large lesions or lesion located in the transition zone).In addition, I suggest providing a more detailed description of baseline patients characteristics (prostate volume).
Discussion. The authors should discuss the potential difficulties associated with the presence of coexisting prostatic disease – e.g. granulomatosis prostatis .
Please discuss how the COVID-19 global emergency may affect application of MRI-based techniques in clinical practice ( Please discuss: https://pubmed.ncbi.nlm.nih.gov/33904992/; https://pubmed.ncbi.nlm.nih.gov/33070961/; https://pubmed.ncbi.nlm.nih.gov/32570240/)
Reviewer 3 Report
Summary: In this paper, the authors compare deep learning and radiomics based machine learning classifiers in the context of distinguishing clinically significant and insignificant prostate cancer using multi-parametric MRI. They trained the model on 1 cohort and validated on 3 external cohorts. They observed that while training performance was largely similar, the performance on external cohorts was superior using radiomics compared to deep learning.
The strength of this study is the use of multi institutional cohorts for training and validation. However, there are several concerns in the way the experiments were set up and results evaluated that affect the conclusions made by the authors. These are detailed below:
- Firstly, the problem of comparing deep learning and radiomics is not straightforward as there are several radiomics packages and DL architectures - we can't come to a definite conclusion until we compare a few different approaches and explicitly control for the training conditions and splits. The authors have used a previously published DL architecture and an open source radiomics and machine learning package which provide only a specific set of characteristics.
- Previous works comparing DL and radiomics have not been discussed in context. For instance Truhn et. Al. Radiology 2018.
- The input presented to DL and radiomics must be consistent. DL has access to the entire image while radiomics uses the lesion ROIs. For instance, Hiremath et. Al. Lancet Digital Health 2021 used lesion ROIs as input to the DL architecture which is closer to the radiomics input.
- The cross-validation approach is very different for both methods. Here DL uses a 3 fold cross validation while radiomics with WORC used a 100x test train split, 5 fold cross validation. This implies that the patients being used in a single fold are different for radiomic and DL method. The split of patients in each fold must be controlled when both approaches are being compared.
- There are clearly a number of inconsistencies among the datasets – a) the ground truth is based off of biopsies and whole mount pathology, b) some of the ROIs are from radiologist while some are from mapping from whole mount pathology, c) the b-values are different between the cohorts, d) there could be reader bias in ROI delineations.
- T2W images are not quantitative and need to be pre-processed using bias field correction, standardization methods. The authors can refer to the IBSI guidelines for pre-processing.
- Another concern is that the DL architecture here focuses on the segmentation problem whereas the radiomics focuses on the classification problem. While segmentation result based on posterior probabilities can be used for classification, it is still not a completely fair comparison. We would suggest the authors to train a standalone DL network that takes the same input and training strategy as radiomics for a fair comparison.
- The performance of DL on the PCMM dataset is very concerning. What could be the potential causes for this? Use of pathology mapped ROIs?
- The evaluation metrics are too simiplistic – a table with accuracy, AUC, sensitivity, specificity, F-1 scores can be presented for further insights.